# Kinetic Analysis of Thermal Decomposition of Polyvinyl Chloride at Various Oxygen Concentrations

**Shuo Yang [1], Yong Wang [2,*] and Pengrui Man [2]**

1   Shaanxi Emergency Rescue and Aviation Forest Protection Center, Xi'an 710054, China; 15591979061@163.com
2   Forensic Science Institute, China People's Police University, Langfang 065000, China; 2022909025@cppu.edu.cn
*   Correspondence: wangyong01@cppu.edu.cn

**Abstract:** PVC plastic products are common combustible substances seen in fires, but their thermal degradation behavior under different oxygen concentrations has not been adequately studied. The thermal degradation behavior of PVC materials in atmospheres with different oxygen concentrations was analyzed via thermogravimetric–Fourier transform infrared spectroscopy (TG-FTIR). The TG results show that the thermal degradation process of PVC under a non-oxygenated atmosphere occurred in two stages, and the activation energies of the two stages were 130–175 KJ mol$^{-1}$ and 230–320 KJ mol$^{-1}$, respectively; under the oxygenated atmosphere, the thermal degradation process occurred in three stages. The activation energies of the three stages were 130–175 KJ mol$^{-1}$, 145–510 KJ mol$^{-1}$ and 75–190 KJ mol$^{-1}$, respectively. And the reaction mechanism of the second stage of thermal degradation was changed from D-ZLT$_3$ to E$_n$ by the higher oxygen concentration. Infrared spectroscopy (FTIR) was used to analyze the pyrolysis process of PVC in the non-oxygenated atmosphere, and the eight major components were as follows, in descending order according to amount released: C-H stretching > HCl > C-Cl stretching > H$_2$O > CO$_2$ > C-H bending > C-H aliphatic bending > CH$_2$. For the reaction of PVC at an oxygen concentration of 7%, the nine major components, in descending order according to amount released, were as follows: CO$_2$ > HCl > H$_2$O > CO > C-H stretching > C-Cl stretching > C-H aliphatic bending > C-H bending > CH$_2$. For PVC reactions at oxygen concentrations of 14% and 21%, the five major components, in descending order according to amount released, were CO$_2$ > HCl > CO > C-Cl stretching > H$_2$O.

**Keywords:** oxygen concentration; polyvinyl chloride; thermal decomposition; evolved gases

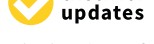



## 1. Introduction

Plastics play an important role in our daily lives and are used to manufacture various items, such as packaging, films, lids, bags and containers. PVC-based plastics are widely used in the manufacture of plastic products such as pipes, window frame materials, rigid films and cables and have excellent resistance to aging and high temperatures [1]. However, in a fire, PVC materials are exposed to high temperatures, and thermal radiation causes a rapid increase in the surface temperature of the material, contributing to pyrolysis and the release of pyrolysis gases. For example, in electrical fires, PVC is often used as an insulation material for wires, and the PVC material undergoes pyrolysis and combustion reactions, releasing large amounts of toxic gases [2]. In fact, PVC products (e.g., cables, pipes, floors, etc.) are often placed in confined spaces or small areas, resulting in a lack of sufficient oxygen on the surface of PVC products. This is likely to affect the pyrolysis and combustion reactions of PVC materials. Under a non-oxygenated atmosphere or low oxygen concentration, the combustion of plastics, wood, etc., will produce more toxic gases, such as CO, due to insufficient oxidation. For PVC materials, a typical polymer plastic, pyrolysis under different oxygen concentrations can produce different and complex pyrolysis products, some of which are toxic and harmful gases that can lead to serious

injuries and economic losses and increase the difficulty of fire suppression. Therefore, it is crucial to study the variation in oxygen concentration in fires to understand the pyrolysis process of PVC materials and the effect of the released gases [3].

In recent years, scientists have studied the pyrolysis process and the pyrolysis products of PVC materials during fires. Thermogravimetric–infrared coupled analysis (TG-FTIR) can be used as the main method to study the pyrolysis process and pyrolysis products of substances under different oxygen concentrations. Benes et al. [4] used thermogravimetric analysis (TGA) combined with mass spectrometry and Fourier transform infrared (FTIR) spectroscopy to study this process. Thermal degradation of VC insulation using specimens was found to occur in three temperature ranges: 200–340 °C, 360–530 °C and 530–770 °C. And the PVC skeleton started to degrade in the range of 200~340 °C with the release of HCl, $H_2O$, $CO_2$ and benzene. Bhargava et al. [5] used the thermogravimetric analysis technique to carry out detailed kinetic analyses of PVC materials using a model-free approach and distributional reactivity models, and various distributional reactivity models were used to explain the thermal decomposition process of PVC polymers. Wang et al. [6] investigated the pyrolysis characteristics of PVC using TG-FTIR and found that the zero-order model and the sixth-order model were responsible for the two PVC sheath pyrolysis stages, respectively. Fourier transform infrared spectroscopy (FTIR) analysis showed that for PVC insulation, the release of seven major components were, in descending order: $CO_2$ > C-H stretching > C-H bending > $H_2O$ > C-H aliphatic bending > $CH_2$ deformation > C-Cl stretching. Wang et al. [7] used multiple thermogravimetric experiments in combination with FTIR analysis to determine the pyrolysis behavior of PVC. Possible reaction mechanisms were predicted using the generalized master plot method, and the relevant pre-exponential factors were calculated using the compensation effect. The analysis yielded changes in thermal degradation behavior attributable to changes in chemical composition, molecular structure, compositional ratios and various additives. In their study on pyrolysis and the combustion of substances under different oxygen concentrations in a fire, Fang et al. [8] investigated the pyrolysis and combustion process of wood under different oxygen concentrations using TG-FTIR analysis and analyzed in-depth the effect of oxygen concentration on the pyrolysis process and fugitive gases. Moreover, the apparent activation energy of pyrolysis and combustion varied linearly with oxygen concentration, and the effect of oxygen concentration on the combustion mechanism of wood was investigated. Currently, thermal analysis of PVC cable insulation materials is usually carried out for an air or nitrogen atmosphere. No extensive research has been carried out on how changes in oxygen concentration affect the thermal degradation properties of PVC materials.

Present-day studies on the pyrolysis of PVC using coupled TG-FTIR techniques at different oxygen concentrations are lacking, and the analyses are not yet complete. Based on previous studies, it is known that the thermogravimetric analysis method can effectively analyze the thermal weight loss process under continuous heating conditions [9–11]. Therefore, the heating rates of 10, 20, 30 and 40 $min^{-1}$ were adopted to simulate the heating process of PVC materials in a real fire. The thermal degradation activation energy and reaction mechanism were analyzed using model-free and model-fitting methods, respectively, and the composition of the released gases was determined via characteristic spectroscopy. The aim of this study was to gain insight into the pyrolysis characteristics of polyvinyl chloride at different oxygen concentrations and determine how the pyrolysis and combustion processes are affected by changes in the oxygen concentration gradient. Semi-quantitative methods of infrared spectroscopy were also used to characterize the temperature dependence of the escaping gases. This provides more accurate data to support improved fire protection measures.

## 2. Materials and Methods

### 2.1. Materials and Sample Preparation

This study used commercially available PVC composition materials. The sample material is used in large quantities in the manufacture of commercial PVC products (such

as plumbing, electrical wiring, etc.) and was provided by Fengtai Polymer Material Co., Ltd. (Weifang, Shandong Province, China), as a reference. The particle diameter was 0.06–0.08 cm. Through oven drying at 80 °C for 12 h, the sample species comprising mostly water were removed, ensuring that the mechanism analysis was not affected by atmospheric water molecules.

### 2.2. TG-FTIR Tests

Thermogravimetric analysis is considered to be the most effective and primary method for evaluating the behavior of heat pyrolysis of substances in a fire scenario and for calculating their kinetic parameters. For this purpose, a thermogravimetric analyzer, the STA4499F5 thermogravimetric analyzer model from NETZSCH-Geratebau GmbH, instrument Co., Ltd. (Selb, Germany), was used. The prepared powder samples of PVC insulation powder were placed in an alumina crucible and weighed using an inbuilt TG balance. The sample mass was controlled at $5 \pm 0.2$ mg, the gas flow rate was 100 mL/min, and the atmosphere was filled with nitrogen and oxygen set at 21%, 14%, 7% and 0% oxygen. The heating rates were set to 10, 20, 30 and 40 Kmin$^{-1}$ and the temperature range was 35–1000 °C to simulate the pyrolysis process of PVC under different oxygen concentrations in a fire scenario. Each test was repeated at least three times to verify the reproducibility of the results. The maximum deviation was less than $\pm 3\%$.

TG-FTIR provides additional information on the emission characteristics of gases. The gaseous substances produced through the programmed heating in the TGA entered the Bruker Invenio S spectrometer via a heat transfer tube that can be heated up to 230 °C to prevent heat dissipation condensation of the decomposition products, thus allowing simultaneous STA-FTIR analysis as well as the combination of quantitative TG analysis and FTIR. FTIR has a spectral range of 650–4000 cm$^{-1}$ with a resolution of 4 and 32 cm$^{-1}$ scans per sample.

### 2.3. Kinetics Analysis

To obtain specific data on the thermal degradation induced by the pyrolysis reaction within the thermogravimetric analysis, with regular increases in temperature with time, the pyrolysis parameters obtained from the thermogravimetric analysis can be evaluated using non-isothermal kinetic methods [12]. The conversion rate of PVC insulation during pyrolysis can be expressed as

$$\alpha = \frac{(m_0 - m_t)}{\left(m_0 - m_f\right)} \tag{1}$$

where $m_0$ = initial mass, $m_t$ = instantaneous mass and $m_f$ = final mass. The rate of transformation of a substance can be expressed as

$$\frac{d\alpha}{dt} = \beta \frac{d\alpha}{dT} = K(T)f(\alpha) \tag{2}$$

where $t$, $T$, $\beta$, $f(x)$ and $K(T)$ represent the time (min), temperature (K), ambient warming rate $= dT/d\alpha$ (Kmin$^{-1}$), the differential transformation function of the reaction mechanism and the rate constant of the sample, respectively, as described by the Arrhenius Law in Equation (3).

$$K(T) = A \exp(-\frac{E}{RT}) \tag{3}$$

where $E$, $A$ and $R$ represent the activation energy required for the reaction (kJ mol$^{-1}$), the pre-exponential factor (s$^{-1}$) and the gas constant (8.314 J mol$^{-1}$ K$^{-1}$), respectively. Combining Equations (2) and (3), with $\alpha = 0$ and T being equal to the starting temperature $T_0$, the integral form $g(\alpha)$ of the mechanism function can be expressed as

$$g(\alpha) = \int_0^\alpha \frac{d\alpha}{f(\alpha)} = \frac{A}{\beta} \int_{T_0}^T \exp(-\frac{E}{RT})dT \tag{4}$$

### 2.3.1. Model-Free Method

The model-free method is a standard approach to estimating the activation energy of composites without knowing or assuming a reaction model [13]. The two commonly used model-free methods chosen for this study were the Kissinger–Akahira–Sunose (KAS) [14] and Friedman (FR) [15], models as shown in Table 1.

**Table 1.** Common model-free functional equations.

| Model-Free Method | Expressions |
|---|---|
| Kissinger–Akahira–Sunose (KAS) | $\ln\left(\frac{\beta}{T^2}\right) = \ln\left(\frac{AR}{Eg(\alpha)}\right) - \frac{E}{RT}$ |
| Friedman (FR) | $\ln\left(\frac{d\alpha}{dt}\right) = \ln\left(\beta\frac{d\alpha}{dT}\right) = \ln[f(\alpha)A] - \frac{E}{RT}$ |

### 2.3.2. Model-Fitting Method

The reaction mechanism can be determined using the Málek method [16] and is expressed as follows

$$Y(\alpha) = \frac{Z(\alpha)}{Z(0.5)} = \frac{f(\alpha)g(\alpha)}{f(0.5)g(0.5)} = \left(\frac{T_\alpha}{T_{0.5}}\right)^2 \frac{(d\alpha/dt)_\alpha}{(d\alpha/dt)_{0.5}} \tag{5}$$

where $(d\alpha/dt)_{0.5}$ and $T_{0.5}$ correspond to the conversion rate and temperature at a conversion rate of 0.5, respectively. The mathematical equations on the left-hand side of the equation are general theoretical models that are constructed. These models can be used to characterize the rates of reactions, substance transformations, etc. As shown on the right-hand side, experimental data are obtained by heating the sample and measuring the change in the rate of reaction. In order to determine the most appropriate model to describe the pyrolysis of the sample, we need to compare the experimental data with the theoretical model. If the model can accurately predict the experimental results, then we can assume that the model is most suitable for describing the pyrolysis process of the sample. Shown in Table 2 [17] is the general theoretical model corresponding to the formula. It is a general theoretical model corresponding to the formula used to describe the mathematical expression of the relationship between the change in the thermal properties of a substance in the process of thermal analysis, such as enthalpy change, heat capacity change and other factors, such as temperature and time.

**Table 2.** Common kinetic mechanism (mode) functions [17].

| Number | Model | Differential Form $f(\alpha)$ | Integral Form $G(\alpha)$ |
|---|---|---|---|
| Diffusion model | | | |
| 1 | 1D diffusion $D_1$ | $-1/2\alpha^{-1}$ | $\alpha^2$ |
| 2 | 2D diffusion–Valensi D-V$_2$ | $[-\ln(1-\alpha)]^{-1}$ | $\alpha + (1-\alpha)\ln(1-\alpha)$ |
| 3 | 3D diffusion–Jande r D-J$_3$ | $6(1-\alpha)^{2/3}[1-(1-\alpha)^{1/3}]^{1/2}$ | $[1-(1-\alpha)^{1/3}]^{1/2}$ |
| 4 | 3D Zhuravlev–Leskin–Tempelman D-ZLT$_3$ | $3/2(1-\alpha)^{4/3}\left[(1-\alpha)^{-1/3}-1\right]^{-1}$ | $\left[(1-\alpha)^{-1/3}-1\right]^2$ |
| Sigmoidal rate equations | | | |
| 5 | Avarami–Erofeev A$_2$ | $1/2(1-\alpha)[-\ln(1-\alpha)]^{-1}$ | $[-\ln(1-\alpha)]^2$ |
| Reaction order models | | | |
| 6 | Second-order chemical reaction F$_2$ | $(1-\alpha)^2$ | $(1-\alpha)^{-1}-1$ |
| Exponent power models | | | |
| 7 | Second-order E$_2$ | $1/2\alpha$ | $-\ln\alpha^2$ |
| Geometrical contraction models | | | |
| 8 | Contracting area R$_2$ | $2(1-\alpha)^{1/2}$ | $1-(1-\alpha)^{1/2}$ |

## 3. Results and Discussion

### 3.1. Thermogravimetric Analysis

In order to investigate the pyrolysis process of PVC materials under different oxygen concentrations in fire conditions, the thermogravimetric loss behavior of the whole process of combustion and pyrolysis of PVC materials under different oxygen concentrations was analyzed by determining the TG-DTG curves of experimental samples with four sets of different oxygen concentrations (0–21%) with the heating rates 10–40 °C min$^{-1}$, as shown in Figures 1 and 2. According to a previous study, the pyrolysis process of PVC usually has two main weight loss peaks [18]. The first weight loss peak is the autocatalytic reaction of the dehydrochlorination of PVC material, which is the main weight loss stage of PVC pyrolysis. The second weight loss peak may be due to the breakage of conjugated polyene chains and interactions within the molecule [19]. And, as mentioned in Figures 1 and 2, due to the participation of different concentrations of oxygen in the reaction, the TG and DTG curve with a high oxygen concentration shifts towards the low-temperature section in the later stage of the reaction. The appearance temperature and peak value of the first weight loss peak of PVC pyrolysis did not change significantly with the increase in oxygen concentration. However, the second weight loss peak changed significantly. For the second weight loss peak, the values were 462, 439, 438 and 438 at 10 °C min$^{-1}$ for 0%, 7%, 14% and 21% oxygen concentrations, respectively, and 476, 448, 449 and 447 at 20 °C min$^{-1}$. The values were 487, 453, 455 and 456 at 30 °C min$^{-1}$. The values were 492, 462, 461 and 459 at 40 °C min$^{-1}$, respectively. It can be observed that the characteristic temperature points associated with the second weight loss peak in PVC pyrolysis shifted to higher temperatures as the heating rate increased. However, the characteristic temperature points of the reaction remained essentially unchanged regardless of variations in oxygen concentrations. It is worth noting that there was a third distinct peak for PVC pyrolysis with oxygen, which was seen at the values 567, 542 and 532 at 10 °C min$^{-1}$ for 7%, 14% and 21% oxygen concentrations, 600, 575 and 552 at 20 °C min$^{-1}$, 632, 587 and 572 at 30 °C min$^{-1}$ and 656, 612 and 585 at 40 °C min$^{-1}$, respectively, and the third peak of PVC pyrolysis was sharper at higher oxygen concentrations.

The characteristic values of the main DTG peaks of PVC pyrolysis at different oxygen concentrations are shown in Table 3. It can be observed that the weight loss process of the dehydrochlorination reaction in the pyrolysis of PVC materials was not affected by the presence of oxygen. This is because oxygen reacts with the dehydrochlorination products to form water and other decomposition products. In this reaction, PVC is dechlorinated to form hydrochloric acid and other products. These products then react with oxygen to form water and other oxidation products. This is used to compensate for the elemental hydrogen and chlorine consumed in the oxidation reaction. As a result, the influence on the TG curve is not significant. Further analysis of this phenomenon will be discussed later. It is worth noting that there was more pyrolysis residue of PVC in the non-oxygenated atmosphere than at other oxygen concentrations. This indicates that substances react more completely when oxygen is involved in the reaction. However, the pyrolysis rate in the non-oxygenated atmosphere was always greater than that at different oxygen concentrations until the end of the reaction. This is because the pyrolysis process relies heavily on intermolecular thermal movement and the breaking of chemical bonds. PVC will crack more readily under anaerobic conditions due to the low intermolecular forces between PVC molecules. However, as the oxygen concentration increases, oxygen begins to participate in the reaction process. This introduces additional chemical reaction channels and energy dissipation pathways that reduce the rate of the dehydrochlorination reaction [20]. The reaction rate is greater in the presence of oxygen, where the pyrolysis of PVC molecules undergoes an oxidation reaction. The hydrocarbons in the molecular chain are oxidized by oxygen to form $CO_2$ and $H_2O$. Therefore, as can be seen in the graph, even a small amount of oxygen intervening in the reaction will change PVC from a fast pyrolysis reaction dominated by the dehydrochlorination reaction to a slower oxidation reaction dominated by oxygen.

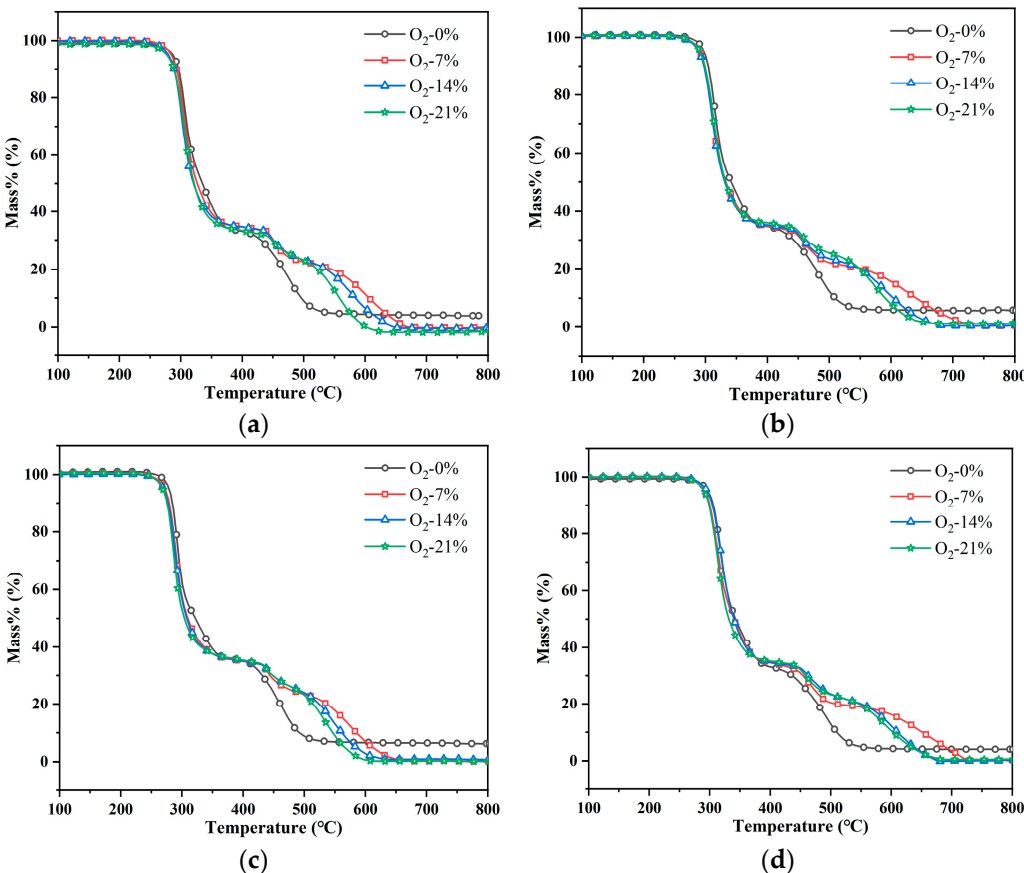

**Figure 1.** Total mass loss (TG) versus temperature curves for PVC with different oxygen concentrations: (**a**) $\beta$ = 10 Kmin$^{-1}$, (**b**) $\beta$ = 20 Kmin$^{-1}$, (**c**) $\beta$ = 30 Kmin$^{-1}$, (**d**) $\beta$ = 40 Kmin$^{-1}$.

**Table 3.** DTG parameters for PVC material under different oxygen concentrations.

| Atmosphere | Heat Rate /°C min$^{-1}$ | $T_{set}$/°C | $T_{final}$/°C | DTG$_{peak1}$ /% min$^{-1}$ | DTG$_{peak2}$ /% min$^{-1}$ | DTG$_{peak3}$ /% min$^{-1}$ |
|---|---|---|---|---|---|---|
| O$_2$-0% | 10 | 275 | 506 | 20.3 | 3.9 | / |
| | 20 | 286 | 523 | 35.7 | 7.5 | / |
| | 30 | 289 | 535 | 53.0 | 10.9 | / |
| | 40 | 292 | 544 | 68.7 | 14.2 | / |
| O$_2$-7% | 10 | 270 | 715 | 19.2 | 3.7 | 2.3 |
| | 20 | 282 | 795 | 36.0 | 7.0 | 4.3 |
| | 30 | 285 | 814 | 52.1 | 8.3 | 4.4 |
| | 40 | 285 | 826 | 63.8 | 12.1 | 5.6 |
| O$_2$-14% | 10 | 268 | 642 | 18.4 | 2.8 | 2.9 |
| | 20 | 279 | 661 | 37.5 | 5.8 | 4.8 |
| | 30 | 283 | 689 | 52.1 | 8.5 | 6.3 |
| | 40 | 289 | 692 | 66.1 | 10.3 | 8.8 |
| O$_2$-21% | 10 | 270 | 610 | 20.0 | 2.9 | 3.27 |
| | 20 | 280 | 636 | 35.0 | 5.0 | 6.6 |
| | 30 | 285 | 663 | 51.4 | 7.5 | 7.9 |
| | 40 | 285 | 740 | 70.5 | 11.7 | 8.4 |

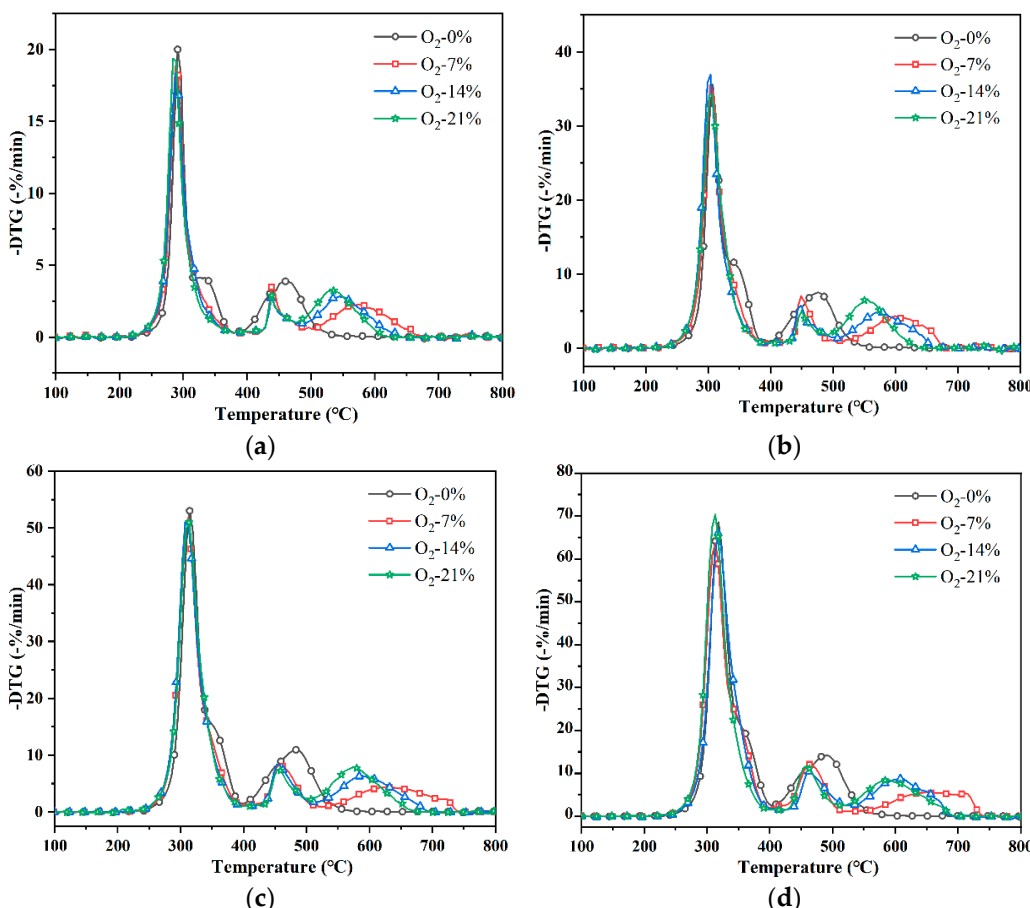

**Figure 2.** Derivative total mass loss (DTG) versus temperature curves for PVC with different oxygen concentrations: (**a**) β = 10 Kmin$^{-1}$, (**b**) β = 20 Kmin$^{-1}$, (**c**) β = 30 Kmin$^{-1}$, (**d**) β = 40 Kmin$^{-1}$.

### 3.2. Activation Energy Analysis

Figure 3 shows the model-free activation energies, E, calculated using the KAS and Friedman methods with different conversion rates (Fitted correlation coefficient R$^2$ values were calculated to be greater than 0.9). It was found that the E values calculated using the KAS and Friedman methods are similar, so they are not discussed separately. And it was clearly found that the activation energy value distribution of the splitting phenomenon, combined with the number of peaks in the previous section, were divided into two or three stages of pyrolysis [21]. In the figure, the splitting point is defined as the point at which there is an abrupt change in the pattern of activation energy change. The splitting point of PVC pyrolysis in the non-oxygenated atmosphere was α = 0.4, and the splitting points of PVC pyrolysis with oxygen participation were α = 0.5 and 0.8. The results show that the pyrolysis of PVC occurs in phases, regardless of whether there is oxygen participation. This is consistent with previous studies [22,23]. Generally speaking, the activation energies at different oxygen concentrations showed a relatively flat temperature trend in the first stage, with activation energy values in the range of 130–175 kJ mol$^{-1}$, while in the second stage, the activation energies showed a wave growth trend due to the complexity and variability of the internal molecular structure reactions in the non-oxygenated atmosphere. In the second stage, due to the complexity of the internal molecular structure under the non-oxygenated atmosphere, the activation energies showed a wave growth trend, and the activation energies were in the range of 230–320 kJ mol$^{-1}$ after a significant decrease at α = 0.7, while the activation energies required for oxidation reactions in pyrolysis reactions with different oxygen concentrations significantly increased and then rapidly decreased. The activation energies were in the range of 145–510 kJ mol$^{-1}$, indicating that the oxidation reaction requires more energy to break the molecular bonds in the pyrolysis

reaction with oxygen. Additionally, in the third stage, the breaking of this part of the bond energy led to the decomposition of the substances that could not be decomposed through pyrolysis. In this case, the hydrocarbon groups in the polymer were decomposed more fully, and the hard-to-decompose bonds, such as aromatic ring bonds, were ruptured to form hydrocarbons and other small groups. This shift in activation energy to the range of 75–190 kJ mol$^{-1}$ signified a significant decrease in the activation energy. It is noteworthy that there was a sudden drop in E in the second stage with no oxygen. The reason for this is that the pyrolysis of PVC is specific due to its own material complexity. Different heating rates affect the reduction in activation energy. This observation is consistent with the TG and DTG curves shown in Figures 1 and 2. It is noteworthy that the change in activation energy with the change in oxygen concentration was not significant. The reason for this may be that the activation energies of the reactants are usually related to the transition state structure and energy of the reaction. In the PVC pyrolysis reaction, with the change in oxygen concentration, the molecular structures of the reactants and products did not change significantly, and the structures of the transition states were relatively stable, resulting in an insignificant change in the activation energy.

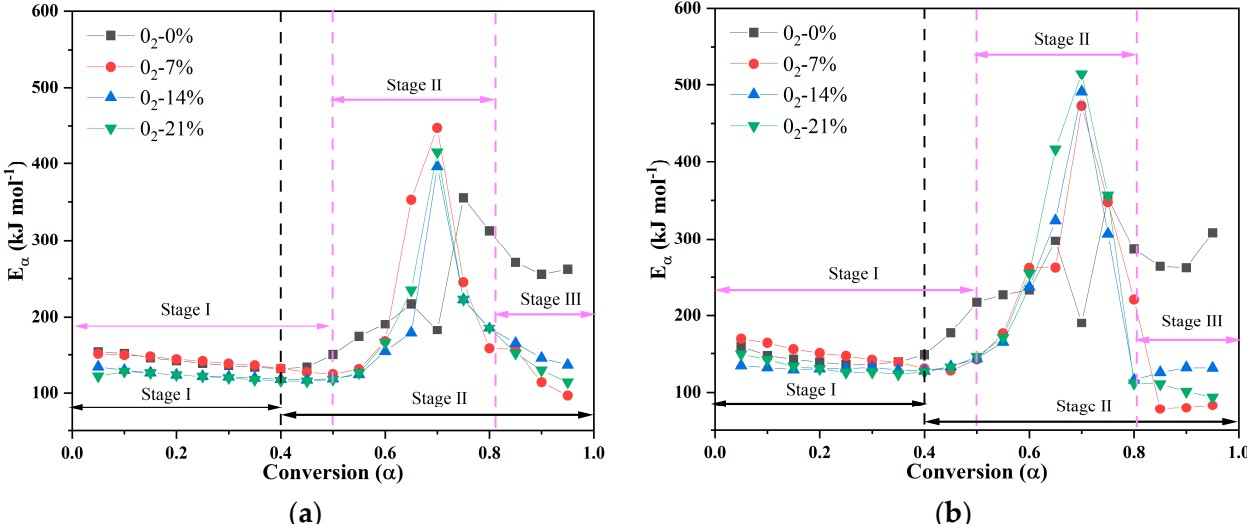

**Figure 3.** Activation energy of thermal degradation of PVC versus conversion α as determined using the (**a**) KAS method and (**b**) Friedman method.

### 3.3. Mechanism Function Analysis

As mentioned above, the pyrolysis process of PVC can be analyzed in two or three stages, and a single model-free approach cannot adequately represent the mechanism function of the PVC pyrolysis process. Therefore, for different stages of the pyrolysis process, the corresponding reaction mechanisms need to be determined [24]. Using the Málek method, experimental and theoretical calculations were plotted, as shown in Figure 4, which shows the experimental (point) and theoretical (line) main plots of the conversion α at different oxygen concentrations. In order to facilitate the comparison of the mechanism functions for different oxygen concentrations, the mechanisms of the second and third stages of the pyrolysis reaction involving oxygen were analyzed together with the second stage of the non-oxygenated atmosphere reaction, as Figure 4b. It is worth noting that the mechanism function of the first stage of PVC pyrolysis did not change significantly after oxygen participation in the reaction. However, the second stage of pyrolysis had a gradual change in the mechanism function due to the increasing concentration of oxygen.

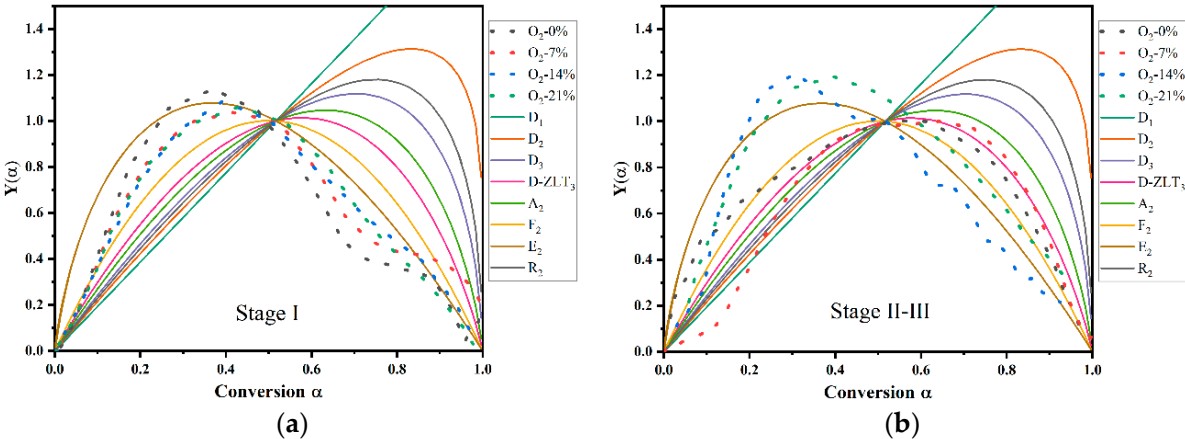

**Figure 4.** Experimental and theoretical master plot results:(**a**) stage I, (**b**) stage II–III.

From the figure, it can be inferred that for the pyrolysis reaction mechanism under the condition of no oxygen, the overall reaction mechanism was most similar to the exponent power models ($E_2$). The reaction mechanisms under the reaction with oxygen participation were all close to the reaction order models $F_2$ ($0 \leq \alpha \leq 0.5$) and $E_2$ ($0.5 \leq \alpha \leq 1$). The changes in the reaction mechanism proved that oxygen participation in PVC pyrolysis changes the decomposition reaction from one dominated by dehydrochlorination in a non-oxygenated atmosphere to an oxidation reaction dominated by oxygen at the mechanistic level. In the second stage, the mechanism function was clearly affected by the oxygen concentration. The mechanism function was closest to that of the 3D Zhuravlev–Leskin–Tempelman (D-ZLT$_3$) model in a non-oxygenated atmosphere and at a 7% oxygen concentration, to $E_2$ at a 14% oxygen concentration and to $F_2$ at a 21% oxygen concentration. These three theoretical models represent the diffusion model, reaction order model and exponent power model, implying that the oxygen concentration changes the dependence between the rate of warming and the reactants. The main reason for this difference is the increased demand for oxygen in the second stage, and a higher oxygen concentration will have more influence on the mechanism of PVC pyrolysis. The higher the concentration of oxygen, the greater the impact on the pyrolysis mechanism of PVC.

From the above discussion, we have gained insights into the thermal cracking behavior of PVC at varying concentrations of oxygen. A simulation of PVC pyrolysis was executed using NETZSCH Kinetics Neo, utilizing the heat loss data calculated earlier (as depicted in Figure 5). Notably, the $R^2$ value of each model's fit exceeded 0.9, underscoring the strong applicability of the fit. The reason why the $R^2$ value at 7% and 14% oxygen was lower than that at 0% and 21% oxygen is because an insufficient oxidation reaction will be affected by the heating rate, which leads to an error in the calculated factor. This, in turn, influences the rate of temperature rise, causing inaccuracies in the computed value of A. In conclusion, NETZSCH Kinetics Neo simulation of the pyrolysis process of PVC under various oxygen concentrations demonstrated its capacity to accurately construct applicable models, thus affirming the reliability of the experiment.

The differences in the pyrolysis process of PVC at different oxygen concentrations were analyzed in detail above. The gases formed during the pyrolysis process can be analyzed via infrared spectroscopy, which provides a scientific analytical method with which to study the quality, nature and quantity of functional groups under different oxygen concentrations [25]. In Figure 6, the three-dimensional TG-FTIR spectrum of the pyrolysis products of PVC under different oxygen concentrations at a heating rate of 40 Kmin$^{-1}$ are shown. Among them, the 3D TG-FTIR spectrum in the non-oxygenated atmosphere was significantly different from the spectrum seen for the pyrolysis reaction involving oxygen. The peak temperatures in the anoxic atmosphere were concentrated at 400–600 °C, while the peak temperatures of the pyrolysis reaction with oxygen were scattered at 350–800 °C. This indicates that the production of fugitive gases from the PVC

pyrolysis reaction involving oxygen was widely distributed in different temperature bands, and the peak of fugitive gases for this reaction was significantly higher than that of fugitive gases in the non-oxygenated atmosphere [26]. However, the peak height did not represent the amount of fugitive gas produced, and the vibration frequencies of different peaks were not the same. The process of fugitive gas production under different oxygen concentrations will be analyzed in detail in the following.

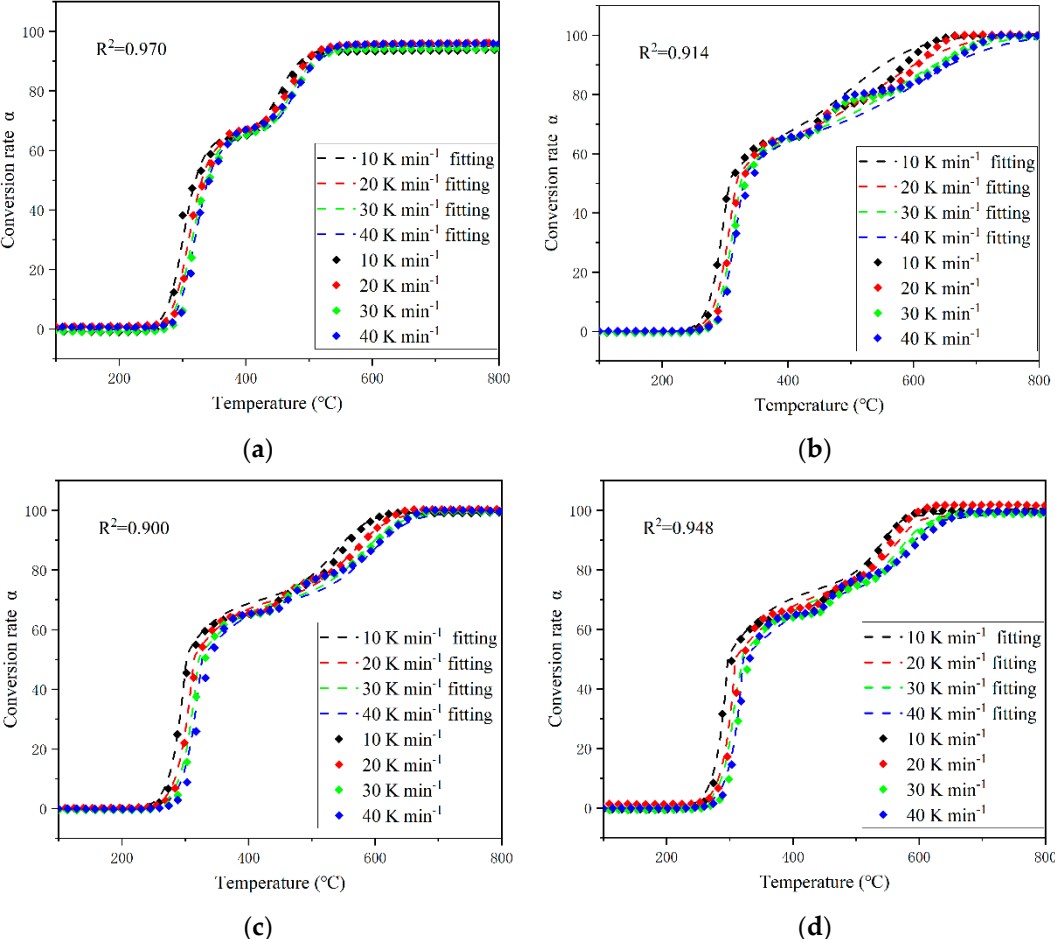

**Figure 5.** Fitted and experimental curves of conversion rate versus temperature at different oxygen concentrations.(**a**) $O_2$-0%, (**b**) $O_2$-7%, (**c**) $O_2$-14%, (**d**) $O_2$-21%.

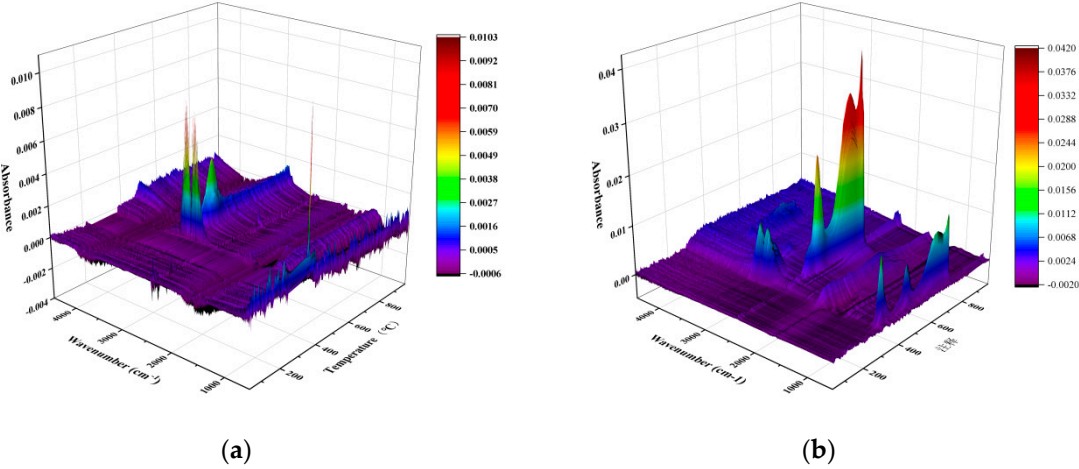

**Figure 6.** *Cont.*

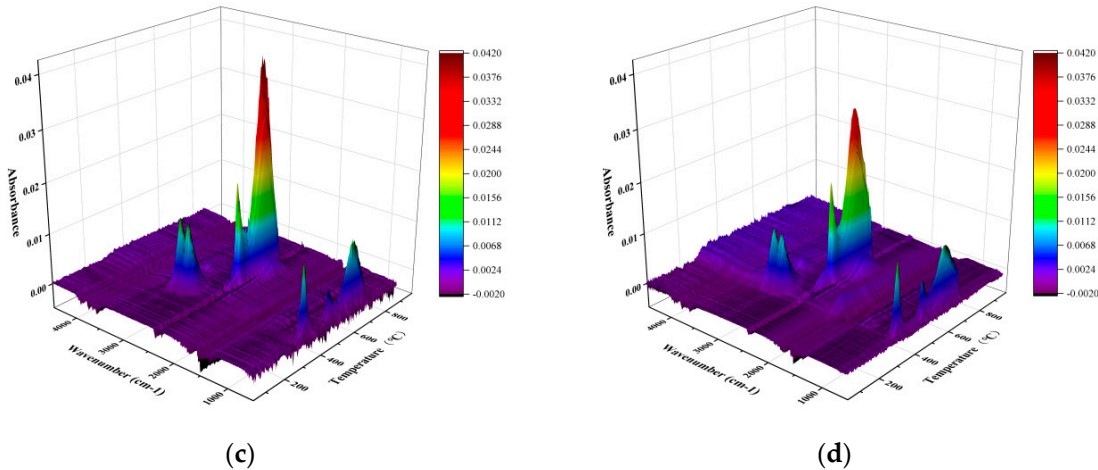

(**c**)    (**d**)

**Figure 6.** Three-dimensional mapping of the FTIR spectrum of the gas at a heating rate of 40 Kmin$^{-1}$: (**a**) $O_2$-0%, (**b**) $O_2$-7%, (**c**) $O_2$-14%, (**d**) $O_2$-21%.

Figure 7 shows the FTIR absorption spectra of the PVC material at a heating rate of 40 Kmin$^{-1}$. Table 4 lists the main vibrational frequency bands of the gases from the PVC material. It can be seen that the main components of the gases were similar for different oxygen concentrations [27]. For the PVC material, there were obvious absorption bands at 4000~3500 cm$^{-1}$ and 1800~1300 cm$^{-1}$, which correspond to O-H stretching vibration and O-H bending vibration, respectively, showing the decomposition of oxygen-containing groups to form $H_2O$. In addition, the characteristic peak at 2934 cm$^{-1}$ indicates the presence of a stretching mode. The characteristic peak at 2934 cm$^{-1}$ indicates the presence of C-H bonds in the stretching mode, which was due to the back-and-forth movement of carbon and hydrogen atoms in the group along the bonding direction. $CO_2$ was located at 2400–2260 cm$^{-1}$, and the peaks of $CO_2$ were very sharp in the reaction involving oxygen. In addition, CO was located at 2173 and 2120 cm$^{-1}$, representing the stretching vibration and bending vibration of the CO molecule. The peak signals for C-H aliphatic bending were weaker near 1458 cm$^{-1}$. As a result of the volume, bond length and bond energy of C-C and C-H bonds in hydrocarbons, they interacted and caused bending of carbon atoms around neighboring carbon–hydrogen bonds. In addition, the characteristic absorption band of $CH_2$ deformation was observed near 1338 cm$^{-1}$. C-H bending near Cl atoms formed an absorption band at about 1240 cm$^{-1}$, which refers to the carbon and hydrogen atoms in the group moving back and forth in a direction perpendicular to the bond. The signals found at 852 cm$^{-1}$ and 660 cm$^{-1}$ can be attributed to C-Cl stretching, which occurred due to a change in the relative plane angle of the C-Cl bond. Combined with the studies summarized above, the main pyrolytic components of the measured PVC insulation in a helium atmosphere were found to be HCl, $H_2O$, $CO_2$, CO, C-H stretching, C-H aliphatic bending, $CH_2$ deformation, C-H bending and C-Cl stretching.

**Table 4.** Band assignments and corresponding wave numbers for PVC.

| Description of Vibrations | Wave Numbers (cm$^{-1}$) |
| --- | --- |
| HCl | 3150–2500 |
| $H_2O$ | 4000–3500, 1800–1300 |
| C-H stretching | 2934 |
| $CO_2$ | 2400–2260 |
| CO | 2173, 2120 |
| aliphatic bending of C-H | 1458 |
| deformation of $CH_2$ | 1338 |
| C-H bending | 1240 |
| stretching of C-Cl | 660, 852 |

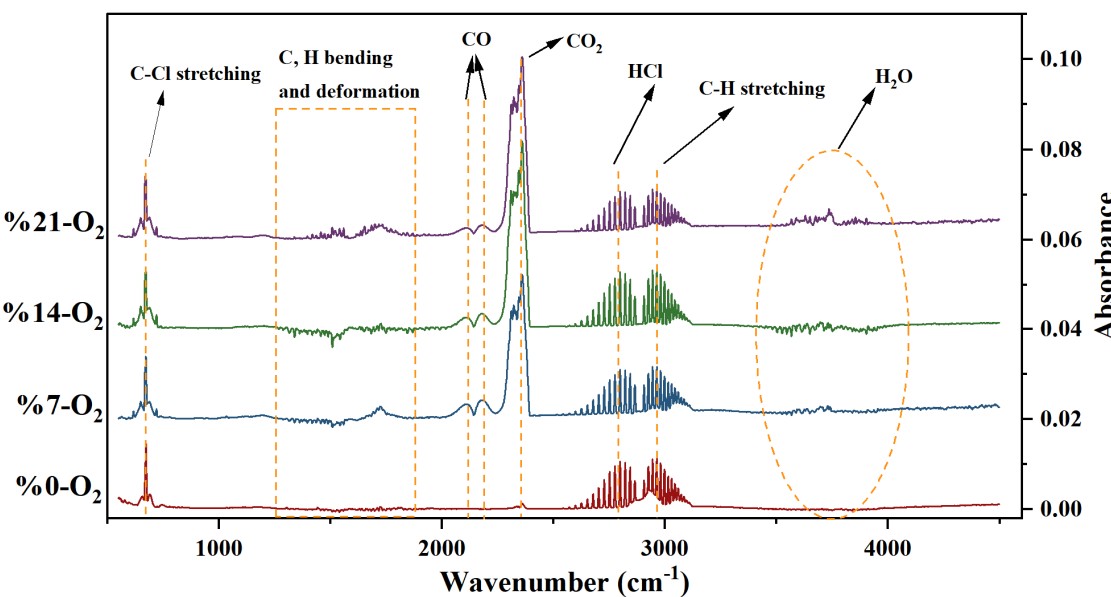

**Figure 7.** Absorption spectra at the heating rate of 40 $Kmin^{-1}$.

Figure 8 shows the regularization curves of the temperature evolution of the main pyrolysis components of PVC at different oxygen concentrations at a temperature increase rate of 40 $Kmin^{-1}$. It was found that the gas of PVC with a temperature increase in the non-oxygenated atmosphere presents two peaks, while the reaction with oxygen participation presents more than two peaks. In PVC pyrolysis, this is the most intuitive phenomenon seen in the oxygen participation reaction regarding the composition of the gas.

For the PVC pyrolysis reaction in the non-oxygenated atmosphere, the first peak, which was mainly a sharp peak formed by the volatilization of HCl gas and the stretching vibration of the C-Cl bond, was in the temperature range of 380–420 °C. The second peak position was in the temperature range of 440–560 °C, and the changes in C-H stretching and vibration, volatilization of the remaining HCl gas, aliphatic bending of the C-H bond, $CH_2$ deformation and C-H bending had a similar trend. For the PVC pyrolysis reaction involving oxygen, the changes brought about by the change in oxygen concentration were obvious. In particular, the change in the $CO_2$ curves at a 7% oxygen concentration showed three peaks and a temperature range of 190–750 °C compared to those at an oxygen concentration above 14%. This is consistent with the weight loss temperature range of the DTG curve. The extra peaks were due to the increased demand for oxygen by the reactants, as evidenced by the increased CO production in the temperature range between the peaks. The most significant changes to occur due to the increasing oxygen concentration were seen in $CO_2$, CO, HCl, C-H stretching and C-Cl stretching, with CO and HCl stretching being the main components of the toxic gases that contribute to the pyrolysis smoke produced by the burning of PVC in fires [28]. Overall, in conjunction with the TG-FTIR analysis process described above, for the pyrolysis process of PVC in the non-oxygenated atmosphere, the seven major components, in descending order according to amount released, were C-H stretching > HCl > C-Cl stretching > $H_2O$ > $CO_2$ > C-H bending > C-H aliphatic bending. The reason for this is that the non-oxygenated atmosphere, as the dominant pyrolysis reaction, leads to weaker C-H bonds, which are easily broken due to high temperatures. Thus, C-H stretching increases the vibration frequency due to the high temperature, releasing a large number of hydrocarbon radicals, which triggers a chain reaction that promotes the decomposition of PVC and the release of other components. For the reaction of PVC at a 7% oxygen concentration, the nine main components, in descending order according to amount released, were $CO_2$ > HCl > $H_2O$ > CO > C-H stretching > C-Cl stretching > C-H aliphatic bending > C-H bending > $CH_2$ deformation. For PVC, the five main components in the reactions at 14% and 21% oxygen concentrations were, in descending order according to amount released, $CO_2$ > HCl > CO > C-Cl stretching > $H_2O$.

The rest of the components did not have significant peaks in the infrared spectrograms due to the difference in oxygen concentration.

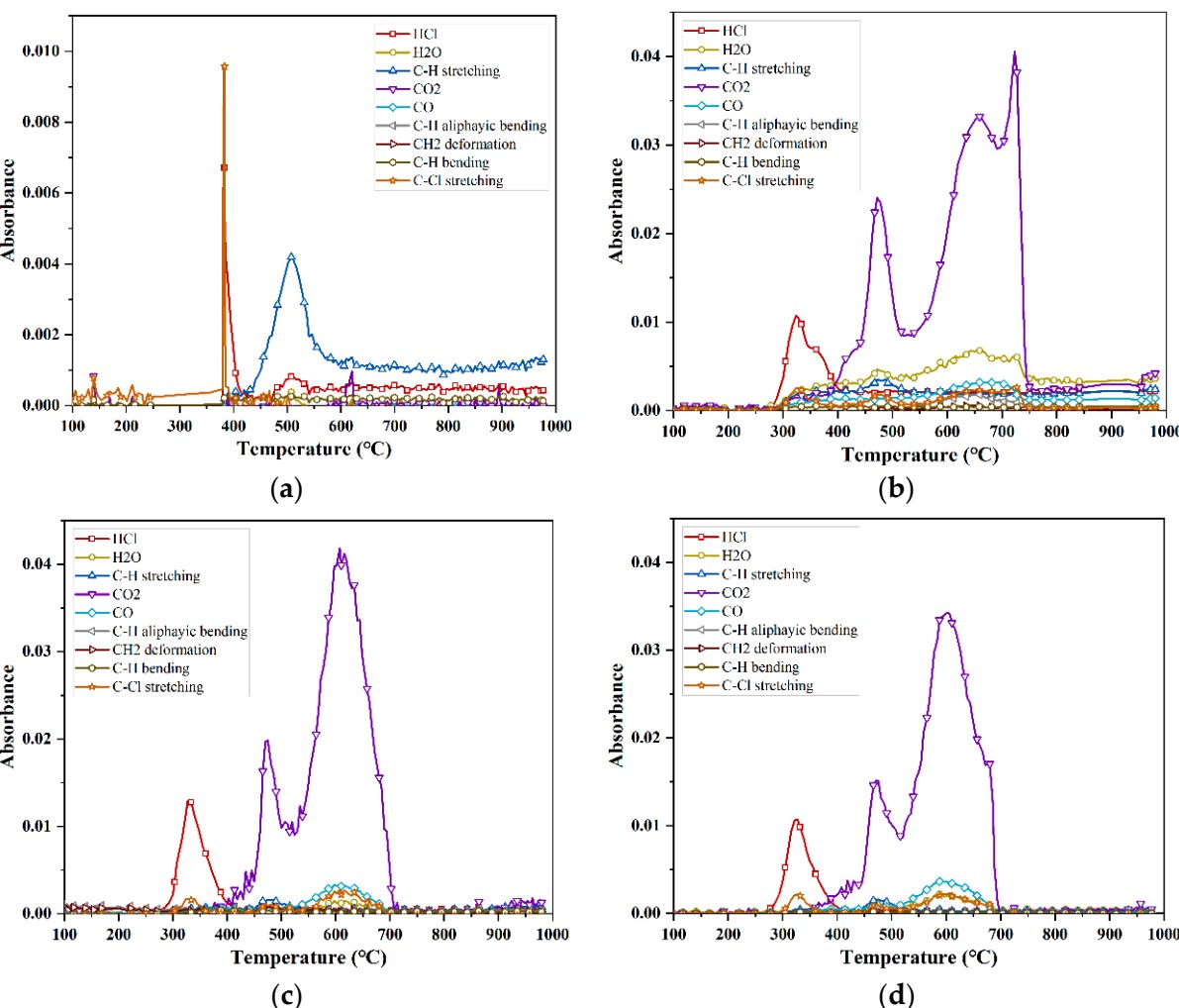

**Figure 8.** Temperature evolution patterns of the main pyrolysis components of PVC: (**a**) $O_2$-0%, (**b**) $O_2$-7%, (**c**) $O_2$-14%, (**d**) $O_2$-21%.

## 4. Conclusions

PVC materials at different oxygen concentrations were analyzed using the thermogravimetric–infrared coupling (TG-FTIR) method in the range of 10–40 $Kmin^{-1}$. The thermogravimetric processes and gas components of PVC materials at different oxygen concentrations were obtained from the analysis. The main conclusions are as follows:

(1)  Different oxygen concentrations and heating rates in a fire affected the pyrolysis process of PVC materials. The increase in heating rate caused the TG curve to shift to the high-temperature section and led to a higher DTG peak. Oxygen concentration mainly affected the second weight loss peak of the PVC pyrolysis process. When oxygen was involved in the pyrolysis reaction, the second weight loss peak changed to two weight loss peaks. The prolongation of pyrolysis and the increase in oxygen concentration led to the gradual acceleration of the oxidation reaction of PVC.

(2)  The activation energies of PVC materials in oxygenated and non-oxygenated atmospheres increased with an increase in the conversion rate in the first stage, and the activation energies were in the range of 130–175 kJ $mol^{-1}$. The activation energies in the oxygenated atmosphere showed a wave-type trend in the first stage, and the activation energies were in the range of 230–320 kJ $mol^{-1}$; the activation energies

in the second stage increased significantly, and then decreased rapidly, in the range of 130–510 kJ mol$^{-1}$. In the third stage, the activation energies rose sharply before falling quickly, ranging from 75 to 510 kJ mol$^{-1}$. The activation energies of PVC in the oxygen-containing atmosphere did not change significantly as a result of the increase in oxygen concentration, and the overall trends of the activation energies were similar.

(3) The effect of oxygen concentration on the mechanism function of PVC pyrolysis was mainly a change in the reaction mechanism of the second pyrolysis stage from D-ZLT$_3$ to E at higher concentrations of oxygen. But there was no significant effect on the reaction mechanism in the first pyrolysis stage.

(4) The pyrolysis of PVC in a non-oxygenated atmosphere was investigated using infrared spectroscopy (FTIR), and the eight major released components, in descending order according to amount released, were C-H stretching > HCl > C-Cl stretching > H$_2$O > CO$_2$ > C-H bending > C-H aliphatic bending > CH$_2$. For PVC at a 7% oxygen concentration, the nine major released components, in descending order according to amount released, were CO$_2$ > HCl > H$_2$O > CO > C-H stretching > C-Cl stretching > C-H aliphatic bending > C-H bending > CH$_2$. For the PVC reaction at 14% and 21% oxygen concentrations, the five major released components, in descending order according to amount released, were: CO$_2$ > HCl > CO > C-Cl stretching > H$_2$O.

**Author Contributions:** Conceptualization, Y.W., writing—original draft preparation, S.Y., writing—review and editing, P.M. All authors have read and agreed to the published version of the manuscript.

**Funding:** This research received no external funding.

**Institutional Review Board Statement:** Not applicable for studies not involving humans or animals.

**Data Availability Statement:** Data is available in a publicly accessible repository.

**Conflicts of Interest:** The authors declare no conflict of interest.

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
