# Peer review of "Kinetic Analysis of Thermal Decomposition of Polyvinyl Chloride at Various Oxygen Concentrations"

_fire, doi:10.3390/fire6100404_

Round 1

Reviewer 1 Report

"Kinetic Analysis of Thermal Decomposition of Polyvinyl Chlo-2ride at Various Oxygen Concentrations" has been reviewed for possible publication in the Fire. The work claimed that they have studied the thermal decomposition of PVC at atmosphere of different oxygen concentration. The author needs to further explain the innovation of this study because there have been many studies on PVC pyrolysis. And there are some issues that need to be addressed before further consideration.

1. In the part of Introduction, the literature review is insufficient to demonstrate the necessity and innovation of studying the thermal decomposition of PVC in an oxygen environment.

2. In Lines 194-198, the authors say that the pyrolysis rate in the no oxygen atmosphere was always greater than that in different oxygen concentration atmosphere. The explanation for this reason should be clearer and more detailed, such as the specific effect of oxygen concentration on the dehydrochlorination reaction.

3. In Lines 215, the number of peaks in the previous section are divided into two or three stages of pyrolysis. Why not differentiate pyrolysis stages in TG and DTG diagrams? What’s the difference?

4.In Line 216, how to obtain the splitting point of PVC pyrolysis?

5.In Figure 3, the Ea undergoes a sudden decrease in Stage II of 0% Oxygen. Why? There is no detailed explanation for this in the text.

6.In 299-233, the detailed explanation the molecular bonds breakage is necessary.

7.In Part 3.3, the analysis here is very unclear, such as results expressed in Figure 4 fail to reveal the corresponding reaction mechanism. The chemical mechanism to be revealed is not clearly expressed. And, the symbols D1, D2, etc. in Figure 4 should correspond to the model in Table 3 and the current identification is not clear.

The grammar and expression errors in the text which should be carefully checked.

Author Response

Thank you for your comments, which are very helpful in improving our manuscripts. We have responded to and corrected each of your comments.

Comments

Reviewer:

Responses to reviewer’s comments

1) In the part of Introduction, the literature review is insufficient to demonstrate the necessity and innovation of studying the thermal decomposition of PVC in an oxygen environment.

The innovation of this paper is as we wrote in the introduction that pyrolysis and combustion of PVC materials produce complex and harmful gases. We investigated how the pyrolysis and combustion processes are affected by a gradient change in oxygen concentration. And we use semi-quantitative methods of infrared spectroscopy to determine the characteristics of the escaping gases as a function of temperature. We will describe our innovations in detail in the introduction.

2)  In Lines 194-198, the authors say that the pyrolysis rate in the no oxygen atmosphere was always greater than that in different oxygen concentration atmosphere. The explanation for this reason should be clearer and more detailed, such as the specific effect of oxygen concentration on the dehydrochlorination reaction.

In response to your comment, we add the following explanation:"The pyrolysis process relies heavily on intermolecular thermal movement and the breaking of chemical bonds. PVC will crack more readily under anaerobic conditions due to the low intermolecular forces between PVC molecules. However, as the oxygen concentration increases, oxygen begins to participate in the reaction process. This introduces additional chemical reaction channels and energy dissipation pathways that reduce the rate of the dehydrochlorination reaction."

3) In Lines 215, the number of peaks in the previous section are divided into two or three stages of pyrolysis. Why not differentiate pyrolysis stages in TG and DTG diagrams? What’s the difference?

The Ictac Kinetics Committee Recommendations for Analysis of Multi-Step Kinetics states. Stages of pyrolysis are generally based on the distribution of the activation energy. If the distribution of activation energy is not obvious, the peak interval of DTG and DSC curves can be used to classify the stages of pyrolysis.The activation energy of PVC pyrolysis process is more clearly segmented, so the activation energy is chosen to distinguish the stages of pyrolysis. It is also more meaningful to show the stage division in the activation energy analysis.

4) In Line 216, how to obtain the splitting point of PVC pyrolysis?

Our definition of the splitting point is the point where there is a sudden change in the pattern of change of the activation energy. In the figure, it is clear that the activation energy change is smooth at the initial stage of pyrolysis or combustion. And the subsequent stages have obvious changes.

5) In Figure 3, the Ea undergoes a sudden decrease in Stage II of 0% Oxygen. Why? There is no detailed explanation for this in the text.

According to previous research, PVC pyrolysis because of its own material complexity peculiarities. The second stage is often in a state of confusion. Therefore the decrease in activation energy here may be due to the different heating rates affecting the decrease in activation energy. We will describe this idea in detail in the paper.

6) In 299-233, the detailed explanation the molecular bonds breakage is necessary.

We have improved the description of molecular bonds breakage.

7) In Part 3.3, the analysis here is very unclear, such as results expressed in Figure 4 fail to reveal the corresponding reaction mechanism. The chemical mechanism to be revealed is not clearly expressed. And, the symbols D1, D2, etc. in Figure 4 should correspond to the model in Table 3 and the current identification is not clear.

We have made the following improvements to the analysis in section 3.3.

Correct the modelling correspondence of Figure 4 and Table 3. Detailed description of the chemical mechanism of the change process of the reaction at different oxygen concentrations.

Reviewer 2 Report

Dear author...

please find attached the reviewed paper with my questions and remarks

Author Response

Thank you for your comments, which are very helpful in improving our manuscripts. We have responded to and corrected each of your comments.

In the abstract,"oxygenated atmosphere" is a pyrolysis reaction in which oxygen is involved. And we believe that "no oxygenated atmosphere" is more accurate as a pyrolysis reaction without the participation of oxygen, and we have made a correction.

In the introduction, we have corrected the grammar and expression. The remaining comments are responded to as follows:

1) It is not clear why you inestigatethe problem, is there the anyrelation to real fire conditions?

The problem we address is the process of thermal cracking of PVC materials at different oxygen concentrations and the release products. According to our research, the atmosphere in a real fire is not simply air or no oxygenated atmosphere. Investigating the effect of oxygen concentration on PVC combustion can provide more accurate data for improving fire prevention measures and algorithmic simulations.

2) why these values?(10,20,30, and 40 min-1)

For the problem of choosing the rate of warming. We chose 10,20,30,40because the rate of temperature rise of burning PVC material in a real fire is in this range. And an equal gradient of increasing rate of temperature rise can be more effective to study the change of mechanism. We will add this account in the text.

In the materials and methods, we have corrected the grammar and expression. “0% oxygen" is isoflow nitrogen, and a detailed description of the theoretical model has been added.

In the results and discussion, We have added the expressions that you think are not described in detail, as well as corrected some errors.

In the Conclusions. "high temperature means that the rate of temperature increase will cause the whole (including feature points and feature curves) to be shifted to the right side of the image. And change the description of the major components.

Reviewer 3 Report

Title: “Kinetic Analysis of Thermal Decomposition of Polyvinyl Chloride at Various Oxygen Concentrations”

Journal: Fire

Comments: The actual manuscript processed thermal decomposition kinetics of PVC in different oxygen concentrations, where the actual research topic belongs to the scope of Fire Journal. The chosen reaction system is not new or modified, but pure combustion process for that considered system is relevant for fire investigation’s and simulations. The particular article issues which should be resolved are listed below.

Abstract section: 1) Please clearly indicate the full name of the abbreviation D-ZLT3 (state the meaning?). 2) Explain the listing of released gases like CO2, HCl and/or CO, and indicate the use or procedure of recycling or post-activation.

Introduction section: 1) Introduce the influence of PVC containing metals on the combustion process through literature reports. Compare PVC without and PVC added metals species, affected the combustion kinetics processes.

Results and discussion section: 1) Since that there is a variation of activation energies against conversion for various oxygen concentrations influenced through separated reaction stages, it is necessary to clearly explain how they were extracted or calculated kinetic triplet components [Ea, A and f(α)], since that complex reaction mechanism asking for placement the reconstruction of the experimental curves using kinetic triplets determined. This implies use the numerical methods, such as Runge-Kutta optimization method or shuffled complex evolution (SCE) optimization method (applied for all oxygen concentrations observed).

Major revision is recommended.

May be improved the English grammar.

Author Response

Thank you for your comments, which are very helpful in improving our manuscripts. We have responded to and corrected each of your comments.

Reviewer:

Responses to reviewer’s comments

The actual manuscript processed thermal decomposition kinetics of PVC in different oxygen concentrations, where the actual research topic belongs to the scope of Fire Journal. The chosen reaction system is not new or modified, but pure combustion process for that considered system is relevant for fire investigation’s and simulations. The particular article issues which should be resolved are listed below.

Your comments are helpful in improving the quality of the manuscript. As you can see we are focusing on providing a more reliable approach to fire investigation's and simulations.

Abstract section:

 1) Please clearly indicate the full name of the abbreviation D-ZLT3 (state the meaning?).

In response to your comments, we have clarified the full name of the D-ZLT3.

2) Explain the listing of released gases like CO2, HCl and/or CO, and indicate the use or procedure of recycling or post-activation.

We have added an explanation of the list of released gases. Our focus is to study the production of PVC pyrolysis gases in the fire scene to determine the fire danger. And recycling or post-activation we may not consider very comprehensively.

Introduction section:1)Introduce the influence of PVC containing metals on the combustion process through literature reports. Compare PVC without and PVC added metals species, affected the combustion kinetics processes.

We add a description of the literature report.

Results and discussion section:1)Since that there is a variation of activation energies against conversion for various oxygen concentrations influenced through separated reaction stages, it is necessary to clearly explain how they were extracted or calculated kinetic triplet components [Ea, A and f(α)], since that complex reaction mechanism asking for placement the reconstruction of the experimental curves using kinetic triplets determined. This implies use the numerical methods, such as Runge-Kutta optimization method or shuffled complex evolution (SCE) optimization method (applied for all oxygen concentrations observed).

Based on your comments we have added specific algorithms for activation energies to the text, as well as the reconstruction of experimental curves using NETZSCH Kinetics Neo software.

Round 2

Reviewer 1 Report

This paper can be published.

Reviewer 3 Report

Title: Kinetic Analysis of Thermal Decomposition of Polyvinyl Chloride at
Various Oxygen Concentrations
Authors: Shuo Yang, Yong Wang *, Peng Rui Man

After revision, the revised manuscript can be accepted for FIRE.